# Effects of Different Types of Recombinant SARS-CoV-2 Spike Protein on Circulating Monocytes’ Structure

**DOI:** 10.3390/ijms24119373

**Published:** 2023-05-27

**Authors:** Marco Vettori, Francesco Dima, Brandon Michael Henry, Giovanni Carpenè, Matteo Gelati, Giovanni Celegon, Gian Luca Salvagno, Giuseppe Lippi

**Affiliations:** 1Section of Clinical Biochemistry, University of Verona, 37129 Verona, Italy; marco.vettori@studenti.univr.it (M.V.); francesco.dima@aovr.veneto.it (F.D.); giovanni.carpene@studenti.univr.it (G.C.); matteo.gelati@univr.it (M.G.); giovanni.celegon@studenti.univr.it (G.C.); gianluca.salvagno@univr.it (G.L.S.); 2Clinical Laboratory, Division of Nephrology and Hypertension, Cincinnati Children’s Hospital Medical Center, Cincinnati, OH 45229, USA; brandon.henry@cchmc.org

**Keywords:** SARS-CoV-2, COVID-19, monocyte, spike protein, morphology

## Abstract

This study investigated the biological effects on circulating monocytes after challenge with SARS-CoV-2 recombinant spike protein. Whole blood collected from seven ostensibly healthy healthcare workers was incubated for 15 min with 2 and 20 ng/mL final concentration of recombinant spike protein of Ancestral, Alpha, Delta, and Omicron variants. Samples were analyzed with Sysmex XN and DI-60 analyzers. Cellular complexity (i.e., the presence of granules, vacuoles and other cytoplasmic inclusions) increased in all samples challenged with the recombinant spike protein of the Ancestral, Alpha, and Delta variants, but not in those containing Omicron. The cellular content of nucleic acids was constantly decreased in most samples, achieving statistical significance in those containing 20 ng/mL of Alpha and Delta recombinant spike proteins. The heterogeneity of monocyte volumes significantly increased in all samples, achieving statistical significance in those containing 20 ng/mL of recombinant spike protein of the Ancestral, Alpha and Delta variants. The monocyte morphological abnormalities after spike protein challenge included dysmorphia, granulation, intense vacuolization, platelet phagocytosis, development of aberrant nuclei, and cytoplasmic extrusions. The SARS-CoV-2 spike protein triggers important monocyte morphological abnormalities, more evident in cells challenged with recombinant spike protein of the more clinically severe Alpha and Delta variants.

## 1. Introduction

The coronavirus disease 2019 (COVID-19) is the most recent life-threatening infectious disease to have reached pandemic proportions in human history [1]. More than three years after its emergence and with over 7 million attributable deaths, SARS-CoV-2 (severe acute respiratory syndrome coronavirus 2) infections are still raising international concern, such that the World Health Organization (WHO) has reiterated the status of public health emergency [2].

Although some aspects of the intricate pathogenesis of COVID-19 remain unraveled, there is now incontrovertible evidence that SARS-CoV-2 is capable of infecting a vast array of human cells besides type II pneumocytes. This is supported by the evidence that the virus can not only bind to its principal host cell receptor angiotensin-converting enzyme 2 (ACE2) but can also associate with a variety of other proteins present at the cell surface such as heparan sulfate, proteoglycans, phosphatidylserine receptor, neuropilin-1, CD147, C-type lectins, SLC1A5, AXL tyrosine-kinase cellular receptor (AXL), kringle containing transmembrane protein 1 (KREMEN1), asialoglycoprotein receptor-1, low-density lipoprotein receptor class A domain 3 (LDLRAD3), and transmembrane protein 30A (TMEM30A), which may directly promote or foster the process of host cell invasion [3,4]. Additional mechanisms of cell entry have been identified, especially characterizing the latest viral variants belonging to the Omicron sub-lineages and encompassing cathepsin-mediated endocytosis, cell-to-cell diffusion, and internalization of SARS-CoV-2-bearing extracellular microparticles generated by infected cells [5].

Throughout the early history of COVID-19, one fact that clearly emerged through many experimental and clinical studies is that the circulating monocytes play a pivotal role in the pathogenesis of this condition [6]. Succinctly, monocytes are not only recruited to the site of infection where they can further differentiate into tissue macrophages or dendritic cells, but they also directly interplay with SARS-CoV-2 in blood, becoming infected, activated, and stimulated to produce a variety of cytokines and chemokines that may support sustained (either localized or systemic) inflammatory reactions [7], which have been conventionally called the “cytokine storm” [8]. Incidentally, both the number and degree of activation of classical monocytes are consistently elevated in patients with COVID-19, especially those expressing CD169 at their surface, a hallmark of monocyte activation [9], whilst their overall number tends to gradually decrease afterward, as a result of cell injury, degeneration, and death directly caused by the virus or by some of its components [10].

Besides the direct role played by the integral viral particle, several lines of evidence suggest that its spike protein may also directly interplay with human monocytes, triggering their activation and thus promoting, maintaining, or even boosting the strong inflammatory reaction that characterizes the more severe cases of COVID-19 [11]. Nonetheless, since the pathogenicity of SARS-CoV-2 has considerably changed over time, generating a lower clinical burden in the infected individuals, we planned this series of experiments to investigate whether different types of recombinant SARS-CoV-2 spike protein may produce diverse alternations of circulating monocyte morphology.

## 2. Results

The results of this study are summarized in Figure 1, Figure 2 and Figure 3. The values of MO-X, reflecting cellular complexity for the presence of granules, vacuoles, and other cytoplasmic inclusions, increased as compared to the control in all samples where the SARS-CoV-2 recombinant spike protein of the Ancestral, Alpha, and Delta variants was added but not in those containing the Omicron variant, with values being typically higher in the samples spiked with the higher 20 ng/mL final concentration compared to those challenged with 2 ng/mL.

The heterogeneity of the monocytes’ complexity was also found to be consistently increased in all samples spiked with the two concentrations of the four SARS-CoV-2 recombinant spike protein variants, achieving statistical significance in samples containing Alpha and Delta 20 ng/mL recombinant spike proteins (Figure 1).

The values of MO-Y, reflecting the cellular content of DNA and RNA, were found to be consistently decreased in most samples spiked with SARS-CoV-2 recombinant spike proteins, achieving statistical significance in those containing Alpha and Delta recombinant spike proteins at 20 ng/mL final concentration (Figure 2).

Unlike MO-Y, MO-WY which reflects the heterogeneity of the monocyte content of DNA and RNA did not vary significantly between the two concentrations of the four different SARS-CoV-2 recombinant spike proteins tested (Figure 2).

As shown in Figure 3, the values of MO-Z, mirroring the presence of monocytes of abnormal size, were found to be significantly decreased in all samples spiked with SARS-CoV-2 recombinant spike proteins (except in the one spiked with 2 ng/mL ancestral variant).

The values of MO-WZ, reflecting the heterogeneity of the monocyte volumes, were found to be increased in all samples, achieving statistical significance in those where the SARS-CoV-2 recombinant spike proteins of the Ancestral, Alpha, and Delta variants were added at the higher 20 ng/mL final concentration (Figure 3).

Table 1 shows a non-significant trend towards larger changes in the samples spiked with the higher 20 ng/mL final concentration of spike protein compared to those challenged with 2 ng/mL, except for MO-WX in the samples challenged with the Omicron spike protein, MO-Y in the samples challenged with the Alpha, Delta, and Omicron spike proteins, and the MO-WY in the samples challenged with the Ancestral spike protein.

The revision of the blood smear performed with the DI-60 digital morphology analyzer confirmed the presence of a broad spectrum of morphological monocyte abnormalities, as shown in Figure 4, which represents some monocytes in the whole sample where the SARS-CoV-2 recombinant spike protein of the Delta variant was added at the final concentration of 20 ng/mL.

Briefly, the cells were found to be largely dysmorphic, granulated, and intensely vacuolated, with an aberrant nucleus and clear alterations in chromatin structure, as well as presenting cytoplasmic extrusions. Platelet phagocytosis could also be clearly observed.

## 3. Discussion

The results of our experimental study, where human monocytes were challenged with different types (Ancestral, Alpha, Delta, and Omicron) and concentrations (2 and 20 ng/mL) of the SARS-CoV-2 recombinant spike protein reveal that the morphology of these cells is dramatically perturbed upon incubation. More precisely, we found increased monocyte complexity (i.e., MO-X), which reflects the presence of granules, vacuoles, and other cytoplasmic inclusions, whilst the monocyte content of DNA and RNA (MO-Y), as well as their size (MO-Z), appeared to be consistently reduced. These changes were also mirrored by a trend towards increased heterogeneity of monocyte complexity and size (MO-WX and MO-WZ, respectively), whilst the heterogeneity of DNA and RNA varied less markedly. Notably, these changes appeared to be consistently dose-dependent, in that the changes displayed a trend, often non-significant, towards larger changes in samples treated with the higher 20 ng/mL final concentration compared to those challenged with 2 ng/mL of the SARS-CoV-2 recombinant spike protein. Importantly, most of these abnormalities found a reliable reflection in the digital images, where we could see that samples with the most evident variations in monocyte parameters displayed clear alterations in monocyte morphology, namely the presence of dysmorphia, increased content of granules and vacuoles, nucleus abnormalities reflecting a considerable alteration of chromatin structure, and even platelet phagocytosis. All these findings provide convincing support different from the biological and epidemiological information previously published and reviewed in the following parts of this article.

The first important aspect that emerged from our study is that the recombinant spike protein of SARS-CoV-2 is capable of strongly disrupting monocyte structure, thus supporting earlier evidence of such intriguing interaction between the cell and the virus. Resting monocytes harbor an abundant concentration of ACE2 and transmembrane serine protease 2 (TMPRSS2) in their cytoplasm, which upon stimulation, can be rapidly transported to the cell surface, where these two proteins are used by SARS-CoV-2 for binding to and fusing with the cell through its spike protein [12]. Virus–cell interaction may also be mediated by the myeloid-cell-specific I-type lectin CD169, to which the spike protein can efficiently bind [13].

In the seminal study published by Ait-Belkacem et al. [11], the authors found that incubation of the SARS-CoV-2 spike protein with human monocytes was capable of triggering a substantial release of inflammatory biomarkers. Important evidence of monocyte–macrophage lineage disruption was reported in an interesting study published by Cao et al. [14], whereby the authors found that spike-protein-pseudotyped lentiviruses were capable of binding to these cell lines, triggering sustained pro-inflammatory cytokines production in the lungs, which could be abolished by deletion of the sequence of the receptor binding domain (RBD) from the spike protein. In a subsequent study, other authors found that the spike protein could promote monocyte activation per se to an extent similar to that seen using other potent monocyte activators such as lipopolysaccharides [15]. Interestingly, enhanced content of lysosomal ionized calcium (Ca^2+^) was also observed, which is an important marker of cell death processes including apoptosis and autophagy [16,17]. A similar effect (i.e., monocyte apoptosis) was observed in a separate experiment by Chandrasekar et al., who challenged human monocytes isolated from the blood of ostensibly healthy individuals with SARS-CoV-2 recombinant spike protein [18] and reported that such incubation with the spike protein triggered caspase 3 activation and subsequent cell death.

The concept that the SARS-CoV-2 spike protein alone could be capable of binding and altering monocyte biology has been convincingly confirmed in a series of experiments conducted by Schroeder et al. [19]. In brief, the authors immobilized different spike protein components within microtiter wells, showing that its S1 subunit effectively bound and activated human monocytes, generating a proinflammatory state characterized by a pattern of cytokine release virtually identical to that seen in patients with COVID-19-induced cytokine storm. In a subsequent study, Barhoumi et al. challenged THP-1 cells (i.e., derived from a human monocytic cell line) with the SARS-CoV-2 recombinant spike protein [20], and found significant induction of apoptosis, with cell activity considerably dysregulated towards the production of reactive oxygen species (ROS). A similar effect was noted using peripheral blood mononuclear cells. Notably, all such apoptotic effects triggered by the SARS-CoV-2 spike protein in other studies are thoughtfully mirrored by our experimental findings. In fact, we also found that adding SARS-CoV-2 recombinant spike protein to human blood samples caused a significant reduction in monocytes’ fluorescence intensity (a reduced MO-Y indicates a lower cellular amount of DNA and RNA, characterizing apoptotic cells) (Figure 3) and the appearance of dysmorphic and intensely vacuolated monocytes in the peripheral blood smear (Figure 4). Both these events reflect a degenerative process, which is a predictable anticipation of cell death. Accordingly, a recent meta-analysis concluded that monocyte counts were significantly lower in COVID-19 patients with severe illness compared to those with milder symptoms (standardized mean difference: −0.56; 95%CI, from −0.8 to −0.32; *p* < 0.001) [21], which is in keeping with our findings that monocytes may be seriously injured after being infected by SARS-CoV-2 or even by interacting with its spike protein alone.

Concerning the morphological abnormalities triggered by monocyte–spike protein interaction that were observed in our study, these changes seem to have similar counterparts ex vivo. An interesting study published by Utrero-Rico et al. assayed circulating monocytes upon emergency department admission in 131 patients with acute SARS-CoV-2 infection [22], reporting the presence of characteristic dysmorphia and intense granularity, with enhanced propensity to release pro-inflammatory cytokines. Volumetric monocyte alternations were clearly seen ex vivo in the study of Kubánková et al. [23], who reported a significant change in the monocyte size of patients with active COVID-19 compared to a healthy control population. In keeping with our observations, the authors also found an increased SD of the cross-sectional monocyte area in patients with COVID-19, which is reflected by the enhanced MO-WZ value that we found in samples where SARS-CoV-2 recombinant spike proteins (except that of the Omicron variant) were added at the final concentration of 20 ng/mL, clearly reflecting a broader heterogeneity of monocyte volumes. Not surprisingly, though, Ropa et al. carried out flow cytometry analyses of blood cells challenged with the SARS-CoV-2 spike protein and found a significantly decreased monocyte size and increased granularity, which are actually consistent with our findings [24]. In another study, Zhang et al. used flow cytometry for studying peripheral blood samples collected from 34 patients with COVID-19 [25]. These authors observed substantial morphological and functional differences in monocytes of COVID-19 patients, which appeared even more evident in those with severe SARS-CoV-2 infection needing prolonged hospitalization and/or intensive care. These authors also reported a remarkably increased number of atypical and vacuolated monocytes in COVID-19 patients, which could not be seen in the blood of healthy individuals. Exceptionally, the ratio between these atypical to typical monocytes in the blood comprised between 60–100% in COVID-19 patients compared to 0% in healthy control blood. It is hence not surprising that we found a high degree of heterogeneity of monocyte volumes in our experiments (i.e., the higher MO-WZ) which could be reflected by the increased value of the monocyte distribution width (MDW: i.e., the relative change in the volume of circulating monocytes). Recent studies and meta-analyses have concluded that MDW is not only increased in patients with SARS-CoV-2 infection compared to those without [26], but it is also a significant predictor of unfavorable clinical progression in those with COVID-19 [27,28,29,30]. Importantly, platelet phagocytosis could also be observed especially in specimens challenged with 20 ng/mL of Alpha and Delta recombinant spike proteins (Figure 4), a pathological process that has been previously described in the blood smears of patients with SARS-CoV-2 infection and reflecting intense monocyte activation and haemophagocytic lymphohistiocytosis [31], a frequent condition seen in patients with COVID-19 [32].

Another important aspect that emerged from our experimental study is that the alteration in monocyte structure seen in the human blood samples challenged with different SARS-CoV-2 recombinant spike proteins was substantially dependent upon the concentration and type of recombinant protein. Succinctly, we typically found larger alterations in samples challenged with 20 ng/mL of the recombinant protein as compared to those where the lower 2 ng/mL concentration was added, as well as in those especially challenged with the SARS-CoV-2 recombinant spike proteins of the Alpha and Delta variants. Thus, it is conceivable that the sequence and structure of these two spike protein variants may display a better affinity for some receptors expressed at the monocyte surface and/or may be capable of more intensely triggering monocyte activation and injury compared to the Omicron spike protein, as was earlier conjectured [33]. The lower chemical stability of the Omicron spike protein compared to the spike protein of former variants has also been suggested as an important characteristic that may explain the lower pathogenetic burden of this strain [34].

Notably, the more accentuated reduction in monocyte fluorescence (i.e., the more markedly reduced MO-Y) observed after challenging human whole blood with a higher concentration of the recombinant spike protein of the Alpha and Delta variants is theoretically consistent with a more intense disruption of cell integrity. On the other hand, the fact that all monocyte parameters were less affected in whole blood challenged with the recombinant spike protein of the Omicron variant seemingly reflects the clinical evidence that the clinical spectrum of disease after infection with the sub-lineages belonging to the Omicron family is substantially milder compared to that caused by the previous strains. Several epidemiological studies, recently meta-analyzed by Hu et al. (33 studies, including over 6 million patients with COVID-19) [35], convincingly showed that Omicron infection was associated with a 2.9-fold lower risk of hospitalization, 3.6-fold lower risk of ICU admission, and a cumulatively 3.0-fold lower risk of death compared to the Delta variant in the general population. Esper and colleagues correlated COVID-19 disease severity and outcomes throughout the periods of major variants dominance [36], reporting that the clinical burden (as defined in terms of hospitalization, ICU admission, and mortality) was similar between the Alpha and Delta waves but in both cases was higher than that seen after the emergence of the Omicron variant. A comparative analysis of the clinical severity of COVID-19 cases sustained by the Alpha, Delta, and Omicron SARS-CoV-2 variants has also been conducted by Varea-Jiménez et al. [37], who concluded that infections sustained by the Alpha and especially Delta variants were associated with higher clinical severity compared to Omicron infections, irrespective of vaccination status. Even more interestingly, Korobova et al. assayed the concentration of multiple cytokines in COVID-19 patients infected by the four major strains (i.e., Ancestral, Alpha, Delta, and Omicron) [38] and reported that only four of these mediators displayed were constantly increased irrespective of the genetic variant. These authors also reported that the so-called cytokine storm and the general “hypercytokinemia” exhibited marked attenuation after the emergence of the new Omicron variant. Interestingly, these findings were confirmed in a subsequent study published by Barh et al. [39], who reported that pro-inflammatory cytokines production (especially interleukin 6) is decreased in patients infected with the Omicron variant compared to those infected by the Delta variant. In keeping with these findings, Park et al. studied nearly 30,000 veterans with Alpha, Delta, and Omicron infections [40] and observed that the risk of having enhanced levels of C reactive protein was higher in those infected by Delta (odds ratio [oR], 2.30; 95%CI, 2.18–2.4) or Alpha (OR, 2.29; 95%CI, 2.14–2.45) compared to Omicron. A similar trend was seen by measuring ferritin (i.e., another well-known inflammatory biomarker), in that the likelihood of having increased concentrations of this protein was higher in COVID-19 patients infected by Delta (OR, 1.91; 95%CI, 1.81–2.02) or Alpha (OR, 1.54; 95%CI, 1.45–1.65) compared to Omicron. Finally, the International Kawasaki Disease Registry reported that the clinical burden of multisystem inflammatory syndrome in children (MIS-C), i.e., a delayed hyperinflammatory response to SARS-CoV-2 infection in children, was found to be considerably lower during the Omicron period compared to the previous Ancestral, Alpha, and Delta waves [41].

In summary, the previously described clinical evidence is strongly congruent with our findings, since disruption of classical monocyte biology seems highest after challenge with recombinant spike proteins of the Alpha and Delta variants, intermediate with that of the Ancestral strain, and the lowest with the recombinant spike protein of the new Omicron sublineages. Nonetheless, the still abnormal value of some monocyte parameters (especially MO-Z, i.e., reflecting an increased presence of monocytes of abnormal size) observed when we added the recombinant spike protein of the Omicron variant would suggests that the pandemic may not be over, at least clinically, as monocyte biology and structure may still be disrupted by the descendants of this most recently emerged variant.

## 4. Materials and Methods

Whole venous blood samples were drawn from seven ostensibly healthy healthcare workers (mean age 50 ± 7 years; 5 females) in the early morning into nine consecutive evacuated blood tubes containing 3.2 M buffered sodium citrate. All blood samples were left standing in a vertical position for at least 15 min after collection and then used for the experiments. Specifically, one blood sample was spiked with 11.6 µL saline, while in the other samples, different variants and concentrations of recombinant SARS-CoV-2 trimeric spike proteins were added (ACROBiosystems, Newark, DE, USA; Ancestral spike: Cat. No. SPN-C52H3; Alpha spike: Cat. No. SPN-C52H6; Delta spike: Cat. No. SPN-C52He; Omicron spike: Cat. No. SPN-C52Hz) (Table 2).

The material, lyophilized from 0.22 μm filtered solution in PBS with trehalose and certified by the company, displays over >90% purity as established by sodium dodecyl sulphate—polyacrylamide gel electrophoresis (SDS-PAGE) and size-exclusion chromatography—multi-angle light scattering (SEC-MALS). The endotoxin content was certified to be <1.0 EU/μg, and it is stable in lyophilized form for 12 months at −20 °C to −70 °C. The resuspended protein is also stable for approximately 3 weeks at 4 °C. The lyophilized recombinant SARS-CoV-2 trimetric spike proteins of the Ancestral, Alpha, Delta, and Omicron viral variants were resuspended with saline, producing two stock solutions with concentrations of 6 × 10^6^ and 6 × 10^5^ ng/mL. The remaining eight whole blood anticoagulated samples were then spiked with 11.6 µL of each spike protein variant and with both solutions, achieving final concentrations in the test samples of 2 and 20 ng/mL, which reflects an interval of blood values of the SARS-CoV-2 spike protein that could be usually measured in patients with acute COVID-19 [42]. An interval of 15 min was observed for processing one sample from the previous, thus allowing us to standardize the time of incubation with the recombinant spike protein (5 min) and for concluding each set of experiments from every single donor within 90 min, which is a guarantee of sample stability [43].

After sample spiking, the specimens were then gently mixed and left standing for 5 min in a vertical position allowing for the interaction between the recombinant spike protein and blood cells. After such a period, the samples were tested on a Sysmex XN hematological analyzer (Sysmex Corporation, Kobe, Japan) connected to a DI-60 digital morphology analyzer (Sysmex Corporation, Kobe, Japan) for obtaining the monocyte parameters. The assessment of leukocytes within a flow chamber in this hematological instrumentation encompasses their illumination by means of laser light, followed by measurement of scattered light in all directions, generating accurate information about the fluorescence intensity, complexity, size, and width of dispersion of events scattered on the three axes of the white blood cell (WBC) differential (WDF) channel, as comprehensively described elsewhere [44]. The monocyte parameters reported on the *x*-axis included the monocytes’ cells complexity (MO-X) and the relative width of dispersion of the monocytes’ complexity (MO-WX); the parameters reported on the *y*-axis included monocytes’ fluorescence intensity (MO-Y) and the relative width of dispersion of the monocytes fluorescence intensity (MO-WY), whilst those reported on the *z*-axis encompass the monocytes’ cell size (MO-Z) and the relative width of the dispersion of monocytes’ cell size (MO-WZ). These parameters and their specific clinical significance are summarized in Table 3 [44].

The results of all these experiments were finally reported as the mean ± SD (standard deviation), whilst the significance of variation from the paired control specimen (i.e., only the saline spiked) was assessed with the Mann–Whitney paired test (Analyse-it; Analyse-it Software Ltd., Leeds, UK). All subjects provided informed consent to be included in this study, which was performed in accordance with the Helsinki Declaration and was approved by the Ethics Committee of the Provinces of Verona and Rovigo (2622CESC).

## 5. Conclusions

In this study, we demonstrated that the spike protein of SARS-CoV-2 is capable of triggering per se morphological abnormalities in monocytes. For the first time, we showed that disruption of monocyte structure becomes more evident when these cells are challenged with the recombinant spike proteins of the more clinically severe Alpha and Delta variants, which may hence explain the attenuated pathogenetic and clinical burden seen after emergence of the Omicron lineages. It is also worth noting that the effects observed in our study may not be directly attributable to the interaction of the SARS-CoV-2 spike protein with monocytes, whereby previous studies suggested that activation of TLR may be an important component of this response, so the morphological changes could also be triggered by signaling after spike protein interaction with lymphocytes or other cell types. It would then be interesting to also study the production of Th1, Th2, and Th17 cytokines by human monocytes incubated with both concentrations of the recombinant SARS-CoV-2 spike proteins, which was not possible at our research facility. Further studies could be planned to investigate this important aspect of monocyte biology.

## Figures and Tables

**Figure 1 ijms-24-09373-f001:**
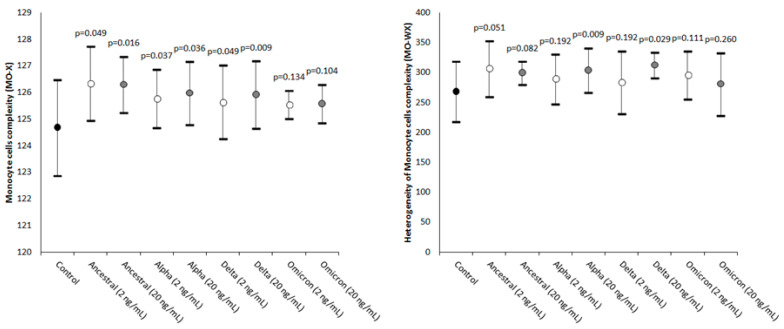
Value distribution of MO-X (monocyte cell complexity) and MO-WX (heterogeneity of monocyte complexity) in human whole blood samples where saline (i.e., control sample) or SARS-CoV-2 recombinant spike proteins of the Ancestral, Alpha, Delta, and Omicron variants were added at a final concentration of 2 and 20 ng/mL. The values are given as the mean and standard deviation (SD); the “*p*” value above the upper limit of the SD indicates the value of statistical significance compared to the control sample where only saline was added.

**Figure 2 ijms-24-09373-f002:**
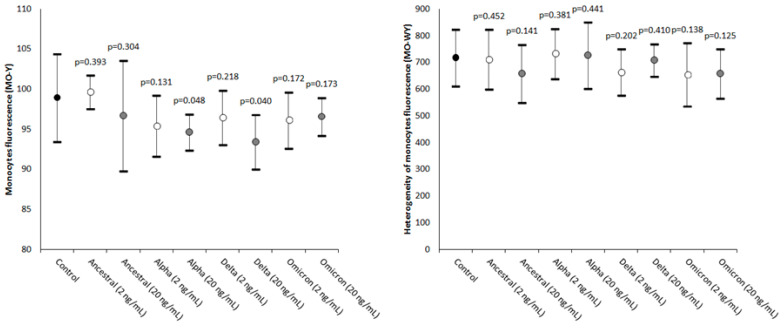
Value distributions of MO-Y (monocytes’ fluorescence intensity) and MO-WY (heterogeneity of the monocytes’ fluorescence intensity) in human whole blood samples where saline (i.e., the control sample) or SARS-CoV-2 recombinant spike proteins of the Ancestral, Alpha, Delta, and Omicron variants were added at a final concentration of 2 and 20 ng/mL. The values are given as the mean and standard deviation (SD); the “*p*” value above the upper limit of the SD indicates the value of statistical significance compared to the control sample where only saline was added.

**Figure 3 ijms-24-09373-f003:**
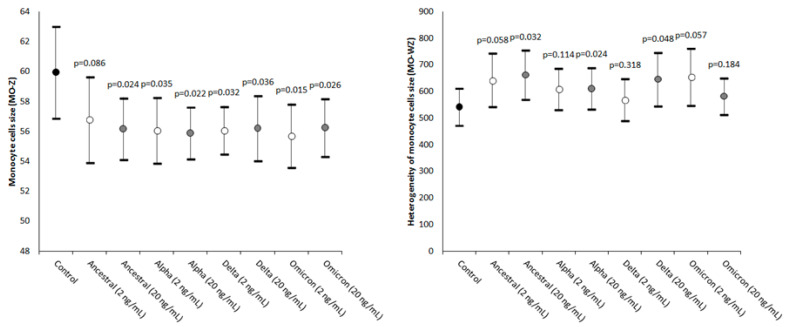
Value distributions of MO-Z (monocyte cells’ size) and MO-WZ (heterogeneity of the monocytes’ cell size) in human whole blood samples where saline (i.e., control sample) or SARS-CoV-2 recombinant spike proteins of the Ancestral, Alpha, Delta, and Omicron variants were added at a final concentration of 2 and 20 ng/mL. The values are given as the mean and standard deviation (SD); the “*p*” value above the upper limit of the SD indicates the value of statistical significance compared to the control sample where only saline was added.

**Figure 4 ijms-24-09373-f004:**
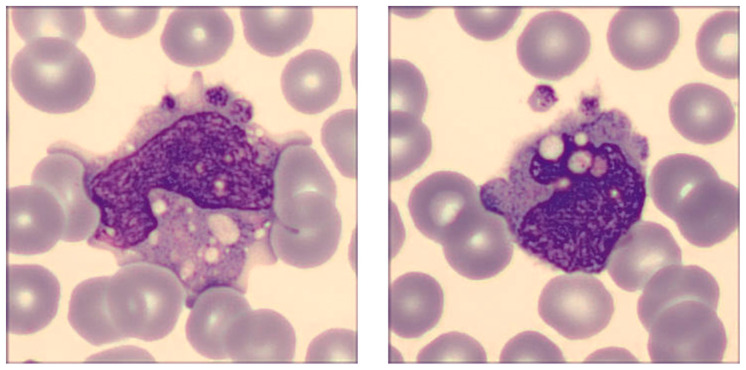
SARS-CoV-2 spike protein-induced alteration of monocyte morphology. The pictures were taken in human whole blood sample incubated for 15 min with 20 ng/mL SARS-CoV-2 spike protein of the Delta variant. Typical distinguishing features encompass dysmorphia, granulation, intense vacuolization, the presence of an aberrant nucleus, and the development of cytoplasmic extrusions. Platelet phagocytosis can also be clearly seen.

**Table 1 ijms-24-09373-t001:** Comparison of monocyte parameters in the samples challenged with 20 vs. 2 ng/mL of the SARS-CoV-2 spike protein.

Parameter	Statistical Significance (20 vs. 2 ng/mL)
MO-X	Ancestral: *p* = 0.467; Alpha: *p* = 0.188; Delta: *p* = 0.386; Omicron: *p* = 0.093
MO-WX	Ancestral: *p* = 0.374; Alpha: *p* = 0.453; Delta: *p* = 0.334; Omicron: *p* = 0.040
MO-Y	Ancestral: *p* = 0.156; Alpha: *p* = 0.008; Delta: *p* < 0.001; Omicron: *p* = 0.003
MO-WY	Ancestral: *p* = 0.049; Alpha: *p* = 0.417; Delta: *p* = 0.470; Omicron: *p* = 0.163
MO-Z	Ancestral: *p* = 0.256; Alpha: *p* = 0.204; Delta: *p* = 0.247; Omicron: *p* = 0.312
MO-WZ	Ancestral: *p* = 0.346; Alpha: *p* = 0.287; Delta: *p* = 0.430; Omicron: *p* = 0.136

**Table 2 ijms-24-09373-t002:** SARS-CoV-2 recombinant spike proteins used for the experiments, as purchased in lyophilized stable and certified form from ACROBiosystems (Newark, DE, USA).

Spike Protein Variants	Additional Mutations
Ancestral	D614G
Alpha (B.1.1.7)	AAVal16, Pro1213, HV69-70del, Y144del, N501Y, A570D, D614G, P681H, T716I, S982A, D1118H
Delta (B.1.617.2)	AAVal16, Pro 1213, T19R, G142D, EF156-157del, R158G, L452R, T478K, D614G, P681R, D950N
Omicron (B.1.1.529)	AAVal16, Pro 1213, A67V, HV69-70del, T95I, G142D, VYY143-145del, N211del, L212I, ins214EPE, G339D, S371L, S373P, S375F, K417N, N440K, G446S, S477N, T478K, E484A, Q493R, G496S, Q498R, N501Y, Y505H, T547K, D614G, H655Y, N679K, P681H, N764K, D796Y, N856K, Q954H, N969K, L981F

**Table 3 ijms-24-09373-t003:** Description of the monocyte parameters obtained using the Sysmex XN hematological analyzer (Sysmex Corporation, Kobe, Japan).

Parameter	Abbreviation	Clinical Significance
Monocyte cells’ complexity	MO-X	Presence of granules, vacuoles, and other cytoplasmic inclusions
Width of dispersion of the monocytes’ complexity	MO-WX	Heterogeneity of monocytes’ complexity
Monocytes’ fluorescence intensity	MO-Y	Quantity of cellular DNA and RNA
Width of dispersion of the monocytes’ fluorescence intensity	MO-WY	Heterogeneity of monocytes’ fluorescence intensity
Monocyte cells’ size	MO-Z	Abnormal sized cells
Width of dispersion of the monocytes’ cell size	MO-WZ	Heterogeneity of monocyte cell size

## Data Availability

Complete data are available from the corresponding author upon reasonable request.

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
