# Peer review of "Effects of Different Types of Recombinant SARS-CoV-2 Spike Protein on Circulating Monocytes’ Structure"

_ijms, 2023, doi:10.3390/ijms24119373_

Round 1
Reviewer 1 Report
General comment
Vettori et al have tested the effects of different types of recombinant SARS-CoV-2 spike protein (WT, alpha, delta and Omicron BA.1) on circulating monocyte morphology and cellular composition. The study provides valuable insights into the biological effects of SARS-CoV-2 recombinant spike protein on human circulating monocytes.
The results indicate that incubating monocytes with WT, alpha and delta recombinant spike protein led to increased cellular complexity, characterized by the presence of granules, vacuoles, and other cytoplasmic inclusions. Another important finding was the significant increase in the heterogeneity of monocyte volumes in all samples challenged with the recombinant spike protein. However, this effect was not observed in samples containing the Omicron variant. Furthermore, the cellular content of nucleic acids consistently decreased in most samples.
Overall, these findings suggest that the SARS-CoV-2 spike protein can induce significant morphological abnormalities in monocytes, and presence of monocytes of abnormal size. Furthermore, the severity of these abnormalities appears to be more pronounced in cells challenged with spike protein from the clinically severe Alpha and Delta variants. This research contributes to our understanding of the biological effects of SARS-CoV-2 spike protein on immune cells.
Minor comments.
1. The authors write “The values of MO-X, reflecting cellular complexity for the presence of granules, vacuoles and other cytoplasmic inclusions, increased as compared to control in all samples where SARS-CoV-2 recombinant spike protein of Ancestral, Alpha and Delta variants were added, but not in those containing the Omicron variant, with values being typically higher in samples spiked with the higher 20 ng/mL final concentration compared to those challenged with 2 ng/mL.” The statement saying, … “with values being typically higher in samples spiked with the higher 20 ng/mL final concentration compared to those challenged with 2 ng/mL” is an over interpretation of data because from Fig. 1 (right panel), both concentrations had fairly similar effects on MO-X (no stats were performed comparing the two concentration to establish statistically significant difference in the means +/- SD). This threat interpretation has been carried forward to discussion. I am not convinced from the data (Fig. 1 – 3) presented in this study that 20 ng/mL of the recombinant spike proteins had significantly greater biological effects compared to 2 ng/mL spike concentration. The errors bars are literally overlapping, clearly indicating no difference in their effects. Can the authors generate p values on the variants they claim, “values being typically higher in samples spiked with the higher 20 ng/mL final concentration compared to those challenged with 2 ng/mL?”
2. The downside of this paper is that it lacks experiments to invest the effect of spike proteins on functionality (e.g., ability to induce production of pro-inflammatory cytokines) of the human monocytes in the absence of COVID-19 infection. The references the authors cite are studies done in patients infected with COVID-19 viruses. It would be interesting to look at the production of Th1, Th2 and Th17 cytokines by human monocytes incubated with 2 ng/mL vs 20 ng/mL of the recombinant spike proteins for 24h, and 72h at 37*C 5% CO2. A whole blood assay can be used to address this question and highlight the potential role of different viral variants in modulating immune responses.
Author Response
Overall, these findings suggest that the SARS-CoV-2 spike protein can induce significant morphological abnormalities in monocytes, and presence of monocytes of abnormal size. Furthermore, the severity of these abnormalities appears to be more pronounced in cells challenged with spike protein from the clinically severe Alpha and Delta variants. This research contributes to our understanding of the biological effects of SARS-CoV-2 spike protein on immune cells.
- We are thankful for these valuable comments on our work. We will do our best to improve the manuscript according the suggestions.
- The authors write “The values of MO-X, reflecting cellular complexity for the presence of granules, vacuoles and other cytoplasmic inclusions, increased as compared to control in all samples where SARS-CoV-2 recombinant spike protein of Ancestral, Alpha and Delta variants were added, but not in those containing the Omicron variant, with values being typically higher in samples spiked with the higher 20 ng/mL final concentration compared to those challenged with 2 ng/mL.” The statement saying, … “with values being typically higher in samples spiked with the higher 20 ng/mL final concentration compared to those challenged with 2 ng/mL” is an over interpretation of data because from Fig. 1 (right panel), both concentrations had fairly similar effects on MO-X (no stats were performed comparing the two concentration to establish statistically significant difference in the means +/- SD). This threat interpretation has been carried forward to discussion. I am not convinced from the data (Fig. 1 – 3) presented in this study that 20 ng/mL of the recombinant spike proteins had significantly greater biological effects compared to 2 ng/mL spike concentration. The errors bars are literally overlapping, clearly indicating no difference in their effects. Can the authors generate p values on the variants they claim, “values being typically higher in samples spiked with the higher 20 ng/mL final concentration compared to those challenged with 2 ng/mL?”
- ANSWER: Thanks for this comment with which we definitely agree. We have performed a new statistics, as specifically requested and the results are in line with those expected by the referee (see new table 3): “Table 3 shows that a non-significant trend towards larger changes is samples spiked with the higher 20 ng/mL final concentration of spike protein compared to those challenged with 2 ng/mL, except for MO-WX in samples challenged with Omicron spike protein, MO-Y in samples challenged with Alpha, Delta and Omicron spike protein, MO-WY in samples challenged with Ancestral spike protein.”. The text of our discussion has completely been re-edited according to the evidence that “Notably, these changes appeared to be consistently dose-dependent, in that the changes displayed a trend, often non-significant, towards larger changes in samples treated with the higher 20 ng/mL final concentration compared to those challenged with 2 ng/mL of SARS-CoV-2 recombinant spike protein”. The entire discussion has been reedited accordingly.
- The downside of this paper is that it lacks experiments to invest the effect of spike proteins on functionality (e.g., ability to induce production of pro-inflammatory cytokines) of the human monocytes in the absence of COVID-19 infection. The references the authors cite are studies done in patients infected with COVID-19 viruses. It would be interesting to look at the production of Th1, Th2 and Th17 cytokines by human monocytes incubated with 2 ng/mL vs 20 ng/mL of the recombinant spike proteins for 24h, and 72h at 37*C 5% CO2. A whole blood assay can be used to address this question and highlight the potential role of different viral variants in modulating immune responses.
- ANSWER: Thanks for this comment. We agree. We have no possibility to perform further studies with these biomarkers, which are unavailable in our research facility and we focused out studies on monocyte morphology, as easily attainable through a common hematological analyzer (this is perhaps the strength of our investigation). However, we acknowledge this comment and we have now listed this aspect as a limitation in our study, as follows: “It would then have been interesting to also study the production of Th1, Th2, and Th17 cytokines by human monocytes incubated with both concentrations of the recombinant SARS-CoV-2 spike proteins, which was not possible at our research facility. Further studies could be planned to investigate this important aspect of monocyte biology.”.
Reviewer 2 Report
This is a straightforward study demonstrating differential effects of spike protein from various SARS-CoV-2 variants on monocyte morphology. This is a small-scale study which provides somewhat limited new information, but does make a small contribution to the field and uses outcomes which have some clinical relevance.
Overall the paper was well-written and presented appropriately. I had some minor comments as follows.
1. Were spike proteins known to be endotoxin-free or low endotoxin? Some previous research into spike protein effects on monocytes/macrophages may have been confounded by LPS contamination of the protein stocks.
2. While not strictly necessary, it would be helpful to label figure panels with A, B, C, etc. to aid in distinguishing panels in the legends. Additionally, a figure title reflecting what is measured (e.g., "Monocyte Complexity" for the left panel of Figure 1) would be useful. Although this information is in the caption, more prominent placement in the figure itself would aid in rapid interpretation of the data.
3. It would also be useful to show individual data points on each figure, especially since the N is so small.
4. It is unclear what from what samples the images in Figure 4 are derived. Are these both Delta-treated cells, or do they differ in treatment in some way?
5. It is important to note in the discussion that the effects seen in this study may not be directly due to spike protein interaction with monocytes, since whole blood cultures were used. Since previous studies suggested TLR activation is a component of this response, it is possible that these monocytes changes are driven by signaling after spike interaction with lymphocytes or another cell type.
Author Response
Overall the paper was well-written and presented appropriately. I had some minor comments as follows.
- We are thankful for these valuable comments on our work. We will do our best to improve the manuscript according the suggestions.
- Were spike proteins known to be endotoxin-free or low endotoxin? Some previous research into spike protein effects on monocytes/macrophages may have been confounded by LPS contamination of the protein stocks.
- ANSWER: Thanks for skilled this comment. The material is certified for not containing endotoxin; specific information about this information can be found in the manufacturer’s website (https://www.acrobiosystems.com/P4491-SARS-CoV-2-Spike-Trimer-Protein-His-Tag-%28B11529Omicron%29-%28MALS-verified%29.html): “Endotoxin: Less than 1.0 EU per μg by the LAL method; Purity: >95% as determined by SDS-PAGE”. This same information has been added to the text of the manuscript (“endotoxin content is certified to be <1.0 EU/μg,”).
- While not strictly necessary, it would be helpful to label figure panels with A, B, C, etc. to aid in distinguishing panels in the legends. Additionally, a figure title reflecting what is measured (e.g., "Monocyte Complexity" for the left panel of Figure 1) would be useful. Although this information is in the caption, more prominent placement in the figure itself would aid in rapid interpretation of the data.
- ANSWER: Thanks for this comment. Good point. Figures relabelled accordingly, replacing the abbreviation with the full term of the monocyte parameter.
- It would also be useful to show individual data points on each figure, especially since the N is so small.
- ANSWER: Thanks for this comment. We initially attempted to do so, but the figures appeared dramatically confusing with all the different dot points through 9 different columns. We prefer to keep the figures without dot points.
- It is unclear what from what samples the images in Figure 4 are derived. Are these both Delta-treated cells, or do they differ in treatment in some way?
- ANSWER: Thanks for this comment. This information has now been provided, as follows: “SARS-CoV-2 spike protein-induced alteration of monocyte morphology. Pictures are taken in human whole blood sample incubated for 15 min with 20 ng/mL SARS-CoV-2 spike protein of the Delta variant.”
- It is important to note in the discussion that the effects seen in this study may not be directly due to spike protein interaction with monocytes, since whole blood cultures were used. Since previous studies suggested TLR activation is a component of this response, it is possible that these monocytes changes are driven by signaling after spike interaction with lymphocytes or another cell type.
- ANSWER: Thanks for this very skilled comment, whose contents have been reported in the manuscript accordingly, as follows. “It is also worth noting that the effects observed in our study may not be directly attributable to the interaction of the SARS-CoV-2 spike protein with monocytes, whereby previous studies suggested that activation of TLR may be an important component of this response, so that the morphological changes could also be triggered by signaling after spike protein interaction with lymphocytes or other cell types”.
Round 2
Reviewer 1 Report
I am satisfied with the authors' argument and revisions.